# Predictive Biomarkers for Immune-Related Endocrinopathies following Immune Checkpoint Inhibitors Treatment

**DOI:** 10.3390/cancers15020375

**Published:** 2023-01-06

**Authors:** Almog Shalit, Panagiotis Sarantis, Evangelos Koustas, Eleni-Myrto Trifylli, Dimitris Matthaios, Michalis V. Karamouzis

**Affiliations:** 1Department of Biological Chemistry, Medical School, National and Kapodistrian University of Athens, 11527 Athens, Greece; 2First Department of Internal Medicine, 417 Army Share Fund Hospital, 11521 Athens, Greece; 3Oncology Department, General Hospital of Rhodes, 85100 Rhodes, Greece

**Keywords:** endocrinopathies, immune checkpoint inhibitors irAEs, biomarkers

## Abstract

**Simple Summary:**

In the last decade, immune checkpoint inhibitors have been incorporated into the array of oncologic treatments used in daily practice, leading to the improved prognosis of many cancers. With their widespread use, clinicians are increasingly confronted with a new category of adverse effects, termed immune-related adverse effects. Endocrine dysfunctions are commonly observed immune-related adverse effects that can be successfully treated if discovered early but have the potential for life-threatening consequences if left undiagnosed. Thus, it is imperative to establish predictive biomarkers that indicate the risk of endocrinopathy occurrence and can guide clinical decisions.

**Abstract:**

In recent years, in the context of the increase in the life expectancy of cancer patients, special attention has been given to immunotherapy and, indeed, to immune checkpoint inhibitors. The use of immune checkpoint inhibitors has increased rapidly, and approximately 40% of cancer patients are eligible for this treatment. Although their impact is valuable on cancer treatment, immune checkpoint inhibitors come with side effects, known as immune-related adverse effects. These can affect many systems, including cutaneous, musculoskeletal, cardiovascular, gastrointestinal, endocrine, neural, and pulmonary systems. In this review, we focus on immune-related endocrinopathies that affect around 10% of all treated patients. Endocrine dysfunctions can manifest as hypophysitis, thyroid dysfunction, hypoparathyroidism, insulin-deficient diabetes mellitus, and primary adrenal insufficiency. Currently, there are multiple ongoing clinical trials that aim to identify possible predictive biomarkers for immune-related adverse effects. The design of those clinical trials relies on collecting a variety of biological specimens (tissue biopsy, blood, plasma, saliva, and stool) at baseline and regular intervals during treatment. In this review, we present the predictive biomarkers (such as antibodies, hormones, cytokines, human leukocyte antigens, and eosinophils) that could potentially be utilized in clinical practice in order to predict adverse effects and manage them appropriately.

## 1. Introduction

Cancer is a multifactorial disease, and despite the improvement of life expectancy with treatments such as surgical removal of the tumor, radiotherapy, and chemotherapy, new treatments are also required for even better outcomes [1]. Since 2011, research has been focused on the development of immunotherapy, with great progress seen in the field of Immune checkpoint inhibitors (ICIs). Immune checkpoint inhibitors act by releasing the immune system from its inhibitory regulatory pathways and upregulating the immune response against tumor cells. The most prominent agents of this category, used in clinical practice, are cytotoxic T lymphocyte antigen 4 (CTLA-4) (ipilimumab), programmed cell death protein 1 (PD-1) (nivolumab, pembrolizumab, cemiplimab), and PD-1 ligand 1 (PD-L1) (avelumab, atezolizumab, and durvalumab) inhibitors [2,3]. Immune checkpoint inhibitors can be used on their own or in conjunction with conventional treatments, such as the ones mentioned above [4,5,6]. The use of ICIs has increased rapidly, and approximately 40% of cancer patients in the United States in 2019 are eligible for this treatment [7]. Their widespread use has resulted in the successful management of a variety of solid tumors.

Despite their valuable contribution to cancer treatment, the increased use of ICI treatments resulted in the emergence of a new type of side effect known as immune-related adverse effects (irAEs) [8]. The severity of irAEs is graded according to the Common Terminology Criteria for Adverse Events (CTCAE) from the US National Cancer Institute (grade 1–5) [9], ranging from mild biochemical abnormalities to death (incidence 0.64%) [8]. The irAEs and the efficacy of ICIs are attributed to the same pathophysiologic mechanism, the activation of the immune system. Suppressing immune tolerance leads to the targeting of normal organs in addition to tumor cells. These adverse effects can affect any organ system, including cutaneous, musculoskeletal, cardiovascular, gastrointestinal, endocrine, neural, and pulmonary systems [10]. Most commonly observed toxicities affect the cutaneous and the gastrointestinal system; however, endocrine toxicities are also commonly reported [11]. Approximately 15% of irAEs need treatment with steroids, 7.5% lead to hospitalization, and 2.5% require treatment discontinuation [12].

In comparison to conventional chemotherapy, the onset of adverse effects is highly unpredictable and variable with ICI therapy. Adverse effects have been reported beginning from a few days following treatment initiation to more than a year after treatment completion. However, the median is estimated approximately at 2–16 weeks from treatment onset [13,14]. As a class, ICI relies on the same basic concept of immune activation in order to treat malignancies. However, each agent has a different target and, subsequently, distinct characteristics. The incidence of irAEs differs depending on the specific ICI agent. It is noted that the overall incidence of severe or life-threatening irAEs (grade ≥ 3) is 10–15% for patients under anti-PD-1 treatment and 20–30% for patients treated with ipilimumab. The incidence increases substantially when a combination of CTLA-4/PD-1 is used (55%) [15]. Furthermore, each ICI agent is correlated with a different frequency and type of immure-related events. Khoja L. et al. (2017) reported that colitis, hypophysitis, and rash were more commonly associated with anti-CTLA-4 treatment, while pneumonitis, hypothyroidism, arthralgia, and vitiligo developed following treatment with anti-PD-1 antibodies [16]. The severity of occurring adverse effects also fluctuates depending on treatment type, with CTLA-4 inhibitors being responsible for more severe presentations [17]. In general, irAEs from anti-CTLA-4 agents are dose-dependent, whereas anti-PD-1 agent toxicities appear to be dose-independent [18,19].

Although there are many reviews regarding irAEs as a whole, we found a gap in data for immune-related endocrinopathies and appropriate predictive biomarkers. Immune-related endocrinopathies affect around 10% of all patients treated with ICIs [20]. Endocrine dysfunctions can manifest as hypophysitis, thyroid dysfunction, hypoparathyroidism, insulin-deficient diabetes mellitus, and primary adrenal insufficiency [21]. Pituitary and thyroid dysfunctions represent the majority of reported cases [22,23]. Some agent-specific associations have been demonstrated in the literature, with hypopituitarism mostly seen following anti-CTLA-4 treatment and thyroid dysfunction being correlated with anti-PD-1 and anti-PD-L1 antibodies [24]. In contrast to other irAEs, most immune-related endocrinopathies can be adequately treated with hormone replacement treatment [21]. However, timely diagnosis is crucial, and an untreated endocrinopathy can lead to life-threatening consequences.

The term biomarker is broad and described as “A defined characteristic that is measured as an indicator of normal biological processes, pathogenic processes or responses to an exposure or intervention” [25]. Therefore, biomarkers can be applied in a wide range of instances in clinical practice and can be subcategorized based on their utility. Although definitions may overlap and some subtypes are not yet unanimously accepted, one existing categorization includes diagnostic, predictive, prognostic, monitoring, pharmacodynamics/response, and safety biomarkers [26]. As the name suggests, predictive biomarkers are used to predict if a certain individual or group of people is more likely to experience a specific event in comparison to similar individuals without the biomarker. In the current review, we present the predictive biomarkers that could potentially signal an increased risk of developing irAEs in the endocrine system. A common obstacle in defining a biomarker is the need to explore its validity and provide substantial evidence for its effectiveness and utility in clinical practice [27]. The scientific community attempts to surpass these difficulties by conducting numerous studies and providing sufficient data to support the relationship between the biomarker and the explored event. In this way, we accomplish reducing the uncertainty about the presence of the observed association. Even so, there is no definitive point of when to pass a studied biomarker from research to clinical practice, which can cause uncertainties about interpreting current data [27].

It is imperative to construct appropriate algorithms in order to predict potential risk, monitor each patient accordingly to the derived risk, and ultimately manage emerging endocrinopathies successfully when needed. For that purpose, prognostic biomarkers for each type of immune-related endocrine dysfunction need to be established. In this review, we present current data on markers that have been associated with the development of endocrine irAEs and could potentially be utilized in clinical practice in order to predict adverse effects and manage them appropriately, resulting in an overall improvement of health care services and patient care.

In the sections below, we list biomarkers that are studied for their predictive utility (pre-treatment and during treatment) of specific immune-related endocrinopathies.

## 2. Thyroiditis

### 2.1. Pre-Treatment

TSH: The significance of TSH as a predictive marker for immune-related thyroiditis has been examined extensively in the recent literature. An observational study consisting of 168 patients with advanced solid malignancies treated with nivolumab was conducted at the National Cancer Center in Tokyo, Japan. Thirty-five patients developed thyroid dysfunction during immunotherapy. The study concluded that elevated levels of TSH (>5 μIU/mL) at baseline were correlated with the development of thyroid dysfunction (OR 7.36, 95% CI 1.66–32.7, *p* < 0.01) [28]. Interestingly, three additional studies reported that higher levels of baseline TSH were correlated with the development of immune-related thyroid dysfunction, irrespective of the established normal range. Specifically, Luongo et al. (2021) and Pollack et al. (2019) observed that baseline levels of TSH were significantly higher in patients that developed hypothyroidism after ICI treatment compared to hyperthyroid and euthyroid patients (*p* < 0.0025, *p* < 0.05, respectively) [29,30], while Brilli et al. (2021) reported higher TSH baseline levels in overt thyroid dysfunction (*p* = 0.003) [31]. Following a ROC curve analysis, each group concluded that a cut-off value of 1.67 mIU/L, 1.72 mUI/L, and 2.19 mIU/L, respectively, is optimal to predict the suggested association. As mentioned, these values are lower than the normal upper limit of TSH (normal range: 0.4–4 mU/L). An additional study reviewed 1246 patients with malignant melanoma receiving ICI treatment. Five hundred and eighteen patients (42%) developed ICI-induced thyroid dysfunction during follow-up [32]. Although a statistically significant association was not observed between baseline TSH and thyroid irAEs overall, patients with higher levels of baseline TSH were at a significantly increased risk of developing ICI-induced overt hypothyroidism (OR 2.33 per mIU/L; 95% CI 1.61–3.33; *p* < 0.001).

Thyroid Autoimmunity (TgAbs and/or TPOAbs): Multiple studies have shown a correlation between the presence of anti-thyroid antibodies before the initiation of immune checkpoint inhibitors and the development of immune-regulated thyroiditis. In a prospective study of 209 patients, of whom 19 developed thyroid dysfunction, the presence of anti-thyroid antibodies was positively associated with the development of thyroid dysfunction in patients treated with pembrolizumab (*p* < 0.01) or nivolumab (*p* < 0.001). Additionally, among the anti-thyroid antibody-positive group, patients with an irregular echo pattern were at greater risk of developing thyroiditis [33]. Similarly, Brilli et al. (2021) tested 63 patients under ICI treatment for anti-thyroid antibody presence before the initiation of treatment [29]. Statistical analysis demonstrated that patients with positive anti-thyroid antibodies at baseline were at a higher risk of developing overt thyroid dysfunction (*p* = 0.01). Contrary to the previous study, no differences were observed in the echosonographic findings. Sakakida et al. (2019) measured the baseline anti-thyroid antibody levels of 122 patients with advanced malignancies who received nivolumab or pembrolizumab [34]. Antibody positivity was significantly more frequent in patients that developed thyroid dysfunction compared to the euthyroid group (13 out of 22 patients with thyroid dysfunction vs. 18 out of 100 euthyroid patients, *p* = 0.0002). The same association was reported by Kobayashi et al. (2018) in a prospective study of 66 patients with malignancies treated with nivolumab (*p* < 0.01) [35]. Toi et al. (2019) reported a statistically significant correlation between anti-thyroid antibodies at baseline and ICI-induced hypothyroidism during their study of pre-existing antibodies and the development of irAEs overall. However, the subgroup with hypothyroidism was small [36]. The relationship between thyroid autoimmunity and thyroid irAEs was further reinforced by Luongo et al. (2021). From a cohort of 96 patients treated with ICIs, antibodies were measured in 43 subjects. Hypothyroidism was observed in 3 out of 8 patients with positive autoantibodies in comparison to 3 of the 35 with negative anti-thyroid antibodies (*p* = 0.0003) [30]. In another study, the correlation of each type of anti-thyroid antibody with disease occurrence was additionally examined. The presence of either antibody (TPOAbs and/or TgAbs) was significantly associated with the development of thyroiditis (OR = 9.19, *p* < 0.01). However, when examining each antibody separately, statistical significance remained only for TgAbs (OR 26.5; 95% CI, 8.18–85.8; *p* < 0.001) [28]. A study from Osorio et al. (2017) studied prospective patients with malignant melanoma under pembrolizumab treatment from the clinical study KEYNOTE-001. The presence of anti-thyroid antibodies was statistically correlated with the development of thyroid dysfunction (*p* < 0.0001). However, the presence of anti-thyroid antibodies was examined collectively, with no differentiation between baseline and during-treatment measurements. Moreover, six out of the seven patients that became positive for antibodies during treatment developed the antibodies at the onset of transient hyperthyroidism [37]. A prospective study of 133 patients with advanced melanoma examined the development of common clinical auto-antibodies and irAEs during treatment with ipilimumab [38]. The data revealed an association between treatment with ipilimumab and the development of any autoantibody (*p* < 0.0001), mainly anti-Tg and anti-TPO antibodies. The appearance of anti-thyroid antibodies was not significantly associated with the development of adverse thyroid effects during ipilimumab treatment. However, a critical correlation was reported during subsequent anti-PD-1 treatment in patients that had previously developed anti-thyroid antibodies under ipilimumab therapy. Specifically, after ipilimumab failure, 61 patients that had not already developed thyroid dysfunction qualified to receive anti-PD-1 treatment. From this subgroup, the development of thyroid dysfunction was observed in 4 out of 9 patients who developed anti-thyroid antibodies during previous ipilimumab treatment and 7 out of 48 patients without antibodies (*p* = 0.04). The statistical analysis suggested that the appearance of thyroid autoimmunity prior to anti-PD-1 treatment was significantly correlated with thyroid dysfunction. A retrospective study was conducted with 122 patients with melanoma receiving immune checkpoint inhibitor (ICI) treatment. TgAb and TPOAb positivity at baseline were 100% specific (31% sensitive) and 97% specific (20% sensitive) for the development of thyroid irAEs, respectively [39]. However, no statistical analysis was performed to establish the association. In conclusion, these studies suggest that there is an association between immune-regulated thyroid dysfunction and pre-existing thyroid autoimmunity in patients that receive ICI treatment. Further studies with larger cohorts are needed in order to examine their utility as predictive markers and establish their use in clinical practice.

Cytokines: The utility of cytokines as biomarkers for thyroid irAEs was examined in a prospective clinical study at the Wakayama Medical University [40]. This particular study observed patients with advanced malignancies under treatment with anti-PD-1 antibody (nivolumab or pembrolizumab), anti-CTLA-4 antibody (ipilimumab), or combination therapy. Thyroid function, anti-thyroid antibodies, as well as cytokine levels were measured before and during the treatment. Thirteen patients developed thyroid dysfunction and were recruited for the study. The control group consisted of 13 patients that did not develop any organ irAEs. The study reported that baseline levels of IL-1β, IL-2, and GM-CSF were significantly higher in patients that developed thyroid dysfunction as opposed to patients that did not (*p* = 0.029, *p* = 0.035, *p* = 0.048, respectively).

### 2.2. During-Treatment

Thyroglobulin (Tg): In the same prospective clinical study that was conducted by Kurimoto et al. (2020), patients that exhibited an early increase in serum thyroglobulin (Tg) following immunotherapy were at increased risk of developing ICI-induced thyroid dysfunction (*p* < 0.05) [40]. Consequently, thyroglobulin is a potential biomarker for immune-mediated thyroiditis. Further studies are required to establish a stronger correlation.

Thyroid Autoimmunity (TgAbs and/or TPOAbs): As mentioned above, thyroid antibodies are valuable markers that can potentially be utilized for the prediction of immune-mediated thyroid dysfunction. In addition to their role during baseline measurements, the appearance or elevation of thyroid antibodies during treatment could also signify the initiation of thyroid dysfunction [39,40,41]. In the phase II clinical trial INSPIRE, blood samples were drawn and analyzed for anti-thyroid antibodies before treatment initiation and before the third cycle (week 7) of treatment with pembrolizumab. Patients with anti-Tg >10.0 IU/mL at pre-cycle 3 were at increased risk of presenting signs of thyroid dysfunction (*p* = 0.024). However, the significance was lost when adjusting for age, gender, ethnicity, and PD-L1 status. Nonetheless, in patients with an anti-Tg titer greater than 10 IU/mL, the collected data suggested that an elevation of anti-Tg titer ≥1.5x from baseline was statistically associated with the development of thyroid dysfunction, even in multivariable models [41]. In the same study, anti-TPO titer at pre-cycle 3 was not correlated with disease occurrence, and the observed association of anti-TPO titer elevation from baseline to pre-cycle 3 did not remain in multivariate analysis. In a separate study, Kurimoto et al. (2020) examined 26 patients retrospectively. The researchers concluded that elevated levels of anti-Tg and/or anti-TPO before the third cycle of ICI treatment were correlated with a higher risk for thyroid toxicity (*p* = 0.012, *p* = 0.048, respectively) [40]. Anti-thyroid antibodies have also been associated with the prediction of specific thyroid dysfunction subtypes. Muir et al. (2022) set out to examine the association between anti-thyroid antibodies and thyroid irAEs. Thyroid irAEs were subcategorized into subclinical thyrotoxicosis, overt thyrotoxicosis, and overt hypothyroidism without preceding thyrotoxicosis. TPOAbs and TgAbs were measured at baseline and at the onset of thyroid irAEs, or 30–60 days after treatment initiation, in patients that remained euthyroid. In patients with overt thyrotoxicosis, TPOAbs and TgAbs titer were significantly increased at the time of onset, compared to baseline (*p* < 0.001). Statistically, a significant increase was not observed in patients that remained euthyroid or developed the other thyroid irAE subtypes [39].

Cytokines: The change in cytokine levels during the course of treatment is another potential biomarker that is being examined for its utility in predicting immune-mediated thyroid dysfunction. Kurimoto et al. (2020) examined the levels of IL-8 and MCP-1 before and 4 weeks after the first ICI treatment (at the time of the third ICI treatment) in their cohort. The study reported that patients who developed thyroid dysfunction presented a statistically significant decrease in these chemokine levels compared to the euthyroid group (*p* < 0.05). In addition, a reduction in G-CSF levels was observed in patients with thyroiditis, which was statistically significant in comparison with patients who did not develop adverse thyroid effects (*p* < 0.05) [40]. Furthermore, Muir et al. (2022) measured the serum IL-6 levels before and after treatment initiation in order to examine a potential correlation with the occurrence of thyroid dysfunction. The study was conducted retrospectively in a cohort of 122 patients with malignant melanoma under treatment with ICI. The data suggested that the elevation of IL-6 levels from baseline was significantly associated with the development of overt hypothyroidism (*p* = 0.03). A similar increase was not observed in patients with overt or subclinical thyrotoxicosis and patients without thyroid dysfunction [39].

## 3. Hypopituitarism

### 3.1. Pre-Treatment

Human leukocyte antigen (HLA): The presence of specific HLA alleles can predispose patients to pituitary irAEs when using immune checkpoint inhibitors [42,43,44]. A retrospective analysis of 11 Japanese patients that developed hypopituitarism during treatment with anti-PD-1 or anti-CTLA-4 suggested a positive association with HLA-DR15 (*p* = 0.0014). Specifically, a significant correlation was found between hypopituitarism and HLA-DRB1*1502 (*p* = 0.0021). HLA-B52 and HLA-Cw12 were found to be additional risk factors. However, the researchers stated that the linkage disequilibrium of HLA-DR15, B52, and Cw12 in the Japanese population poses a limitation in the interpretation of the above findings [42]. Similar results were reported in another study of a Japanese cohort. Kobayashi et al. (2021) conducted a case-control study of 62 patients with advanced malignancies under ICI therapy. Five patients developed hypophysitis, and seventeen exhibited isolated adrenocorticotropic hormone deficiency. HLA-DR15 and HLA-Cw12 were significantly more frequent in the hypophysitis group compared to controls (*p* < 0.05). In patients that developed isolated adrenocorticotropic hormone deficiency, the presence of HLA-DR15 and HLA-Cw12, as well as HLA-DQ7 and HLA-DPw9, was statistically significant (*p* < 0.05) [43]. The utility of HLA alleles as biomarkers for isolated adrenocorticotropic hormone deficiency due to anti-PD1 treatment was also examined by Inaba et al. (2019) [44]. HLA-DRB5*01:02 and HLA-DPB1*09:01 were correlated with adverse effect presentation (*p* = 0.045, *p* = 0.017, respectively). Lastly, HLA-DQA1*01:03 and HLA-DQB1*06:01 frequencies were also significantly higher compared to the control group; however, there is a complete linkage between the two alleles [44].

Baseline anti-pituitary antibodies (APAs): In the control study that was conducted by Kobayashi et al. (2021), the presence of anti-pituitary antibodies before the initiation of treatment was significantly higher in patients that developed isolated adrenocorticotropic hormone deficiency in comparison to the control group. Specifically, 11 out of 17 patients who developed isolated adrenal insufficiency had positive baseline APAs as opposed to 1 patient out of 40 controls (*p* < 0.05). APAs targeted ACTH-secreting and FSH-secreting cells in all patients. In some patients that later developed isolated adrenocorticotropic hormone deficiency, APAs additionally recognized TSH-secreting and LH-secreting cells [43]

Baseline anti-GNAL antibodies: Tahir et al. (2019) [45] conducted their study in two stages in order to discover potential autoantibodies that are associated with the development of irAEs and can be utilized as biomarkers. Concerning the development of pituitary dysfunction, the serum of three patients with ICI-induced hypophysitis was analyzed. Antibodies against guanine nucleotide-binding protein G(olf) subunit alpha (GNAL) and integral membrane protein 2B (ITM2B) were identified as potential markers. In order to confirm their significance, the presence of anti-GNAL and anti-ITM2B was further examined before and after treatment in 20 patients receiving therapy. From the confirmatory group, five patients exhibited hypophysitis, and the rest did not. The data suggested that patients who developed hypophysitis had significantly higher levels of baseline anti-GNAL compared to controls [45]. Thus, the presence of anti-GNAL antibodies before treatment can potentially be a predictive biomarker for the development of ICI-induced hypophysitis.

### 3.2. During Treatment

Anti-pituitary antibodies (APAs): In a case-control study, the serum levels of anti-pituitary antibodies were measured before and after ipilimumab administration. Three out of the four patients that developed hypophysitis became positive for APAs after treatment initiation and before the onset of hypophysis dysfunction. The same was not observed in the control group (0/6). The result was statistically significant (*p* < 0.05) [43]. However, the sample size that was included in the specific analysis is small, and further research is needed in order to report significant data. Kanie et al. (2021) detected antibodies against ACTH-secreting and GH-secreting cells in patients treated with PD-1 or PD-L1 inhibitors that developed hypophysitis [46].

Anti-GNAL: In the study of Tahir et al. (2019), the anti-GNAL antibodies after immunotherapy showed a 1.49-fold increase compared to pre-treatment levels. The fold increase was significantly higher in patients that developed hypophysitis as opposed to controls (*p* < 0.001) [45].

Anti-ITM2B: Tahir et al. (2019) also examined the association of anti-ITM2B antibodies with hypophysitis presentation. The research group observed a 1.7-fold increase from pre-treatment to post-treatment serum samples in patients with hypophysitis. The elevation was significantly greater in patients that developed hypophysis irAEs compared to the control group (*p* < 0.001) [45].

TSH: A retrospective cohort study was conducted in patients with advanced melanoma that received ipilimumab treatment. Of the 46 patients that were enrolled in the study, 9 patients developed hypophysitis following treatment with ipilimumab. Biochemical analysis showed dysfunction in the thyrotroph and corticotroph axis in all patients. The involvement of other axes varied depending on the patient. Lower levels of TSH in pre-cycle 4 serum samples were significantly associated with the development of hypophysitis (*p* = 0.006). The TSH fall was observed at a median of 3.6 weeks before the onset of hypophysitis. Furthermore, a fall in TSH ≥ 80% compared to baseline had 100% sensitivity for detecting patients that will develop hypophysitis in the cohort [47]. However, Siddiqui et al. (2021) did not find a statistically significant difference in TSH levels and TSH change between the hypophysitis group and the controls [48].

TSH index (TSHi), Standardised TSH index (sTSHi): TSH index, an “fT4-adjusted TSH”, has been suggested as a marker for early diagnosis of pituitary dysfunction. By correcting TSH for peripheral fT4-mediated suppression of the pituitary, the true function of thyrotroph cells can be estimated. A retrospective study was conducted on all of the patients that were receiving ICI treatment for advanced melanoma at the Royal Marsden Hospital between 2010 and 2016 [48]. From the initial cohort of 308 patients, 134 were included in the study. Seventeen patients developed hypophysitis. Serum TSH, fT4, and fT3 levels from baseline and before the third cycle of treatment were used to assess their potential as predictive biomarkers for the development of ICI-induced hypophysitis. All patients with hypophysitis that were included in the analysis developed the adverse effect after cycle 3. The pre-cycle 3 values of the TSH index and standardized TSH index were significantly lower in the hypophysitis group compared to controls (*p* < 0.001). However, the ROC analysis determined cut-off levels with low sensitivity and specificity for both values (TSHi: 1.675, sensitivity 76%, specificity: 81%, sTSHi: 1.515 sensitivity: 76%, specificity: 81%). Therefore, although there is a statically significant difference between the hypophysitis and control group, a clinical utility is yet to be demonstrated.

Free T4: In the same study, Siddiqui et al. (2021) reported lower levels of fT4 prior to cycle 3 treatment in patients that exhibited hypophysis dysfunction compared to the control group. The results were statistically significant (*p* < 0.001). Nonetheless, ROC analysis determined a cut-off level of 12.35 pmol/l with a sensitivity of 70% and specificity of 80% [48]. Thus, further assessment should be conducted in order to establish clinical significance.

Eosinophil count (/μL): A retrospective study was conducted at Jichi Medical University Saitama Medical Center in order to evaluate the utility of eosinophils as predictive biomarkers for ICI-induced hypopituitarism [49]. Between 2018 and 2020, 12 patients with renal cell carcinoma under treatment with a combination of nivolumab and ipilimumab were recruited. One patient developed hypophysitis, while the rest presented with isolated adrenocorticotropic hormone deficiency. Serum samples were examined at baseline, at pre-onset, and at the onset of symptoms. The statistical analysis suggested a significant increase in the eosinophil count and eosinophil fraction before the development of hypopituitarism (*p* < 0.05). Moreover, four patients presented with hypereosinophilia, determined as eosinophil count of >500/μL, before adverse-effect-onset, while none had hypereosinophilia at baseline (*p* = 0.015). Consequently, the authors concluded that hypereosinophilia is a potential predictive biomarker for the development of ICI-induced hypopituitarism.

Relative eosinophil count (REC): A retrospective study at Hirosaki University Hospital enrolled all patients that received ICI treatment between 1 September 2014 and 31 January 2021. Of the 19 patients who presented with hypopituitarism, 18 presented with isolated adrenocorticotropic hormone deficiency. In order to examine predictive biomarkers for ICI-induced secondary adrenal insufficiency, 17 patients who developed secondary adrenal insufficiency were selected from this large cohort. Additionally, 22 patients that developed only thyroid irAEs were recruited as the control group [50]. Eosinophil values were examined at baseline, before the onset, and at the time of diagnosis. The study reported that patients with elevated relative eosinophil count were at higher risk of developing secondary adrenal insufficiency (OR: 1.46, 95% CI: 1.04–2.05). The increased relative eosinophil count was observed before the onset of adverse effects. The ROC analysis showed a cut-off of 5.6% as the optimal value in order to predict patients at risk (sensitivity: 41.2%; specificity: 95.5%, area under the curve, 0.72).

Rate of eosinophil count: In the same study, Takayasu et al. (2022) demonstrated that the rate of increase in eosinophil count between baseline and pre-onset was significantly associated with the development of ICI-induced secondary adrenal insufficiency (OR: 1.79, 95% CI: 1.13–2.86). An optimal cut-off value of 1.97 was suggested by ROC analysis (sensitivity: 64.7%, specificity: 72.8%, area under the curve: 0.70) [50].

However, it is imperative to mention that several studies have demonstrated an association between eosinophil count and immune-related adverse effects. As mentioned in our review, Nakamura et al. (2019) reported that baseline levels of absolute eosinophil count (AEC) > 240/μL and relative eosinophil count (REC) 1 month after treatment initiation > 3.2% are positively correlated with the development of endocrine irAEs. In their cohort, 2 out of 14 patients with endocrine irAEs experienced hypopituitarism, but there was no mention of secondary adrenal insufficiency [51]. Eosinophilia has also been reported as a separate immunotherapy-induced adverse effect [52,53]. In a retrospective study that aimed to investigate the correlation between absolute lymphocyte count and irAEs, it was simultaneously observed that the absolute eosinophil count at baseline and 1 month after treatment is significantly associated with ≥grade 2 irAEs [54]. Nonetheless, Krishnan, Tomita, and Roberts-Thomson (2020) reported that patients who developed eosinophilia were at an increased risk of developing irAEs of any grade (*p* = 0.042) [55]. Thus, further study is needed in order to determine if eosinophilia is specifically associated with the development of secondary adrenal insufficiency or should be attributed to the development of irAEs in general.

## 4. Type-1 Diabetes Mellitus

Human leukocyte antigen (HLA): Eight hundred and seventy-one patients with advanced malignancies were recruited from three Japanese medical institutions from 2016 to 2021. Patients received anti-PD-1, anti-PD-L1 antibodies, or a combination of anti-PD-1 and anti-CTLA-4 antibodies. HLA frequencies were compared between the seven patients that developed ICI-induced Type 1 DM and patients that did not develop irAEs, as well as the general Japanese population [56]. The frequency of HLA-DPA1*02:02 and DPB1*05:01 alleles and HLA-DPA1*02: 02-DPB1*05:01 haplotype were significantly higher in patients that presented with T1DM compared to controls (*p* = 0.022, *p* = 0.0027, *p* = 0.0093, respectively) and compared to the Japanese population. The alleles HLA- C*01:02, HLA-DQB1*04:01, and HLA-DRB1*04:05 were also more frequent in patients with ICI-induced T1DM in comparison to the Japanese population. However, the results were not consistent when compared with the control group from the cohort. Four additional patients were examined to confirm the significance of HLA-DRB1*04:05 and HLA-DPB1*05:01. The results demonstrated statistical significance. A second study reviewed the HLA alleles of 23 patients with ICI-induced T1DM from Yale New Haven Hospital and the University of California, San Francisco (UCSF) Medical Center. Patients with HLA-DR4 were at increased risk of developing ICI-induced T1DM compared to U.S. Caucasians and patients with spontaneous T1 DM [57]. Furthermore, several reports mention a relation between the presence of HLA-DRB1*04:05 and HLA-DQB1*04:01 with ICI-induced fulminant T1 DM [58,59,60,61]. Further larger studies are needed to establish a significant association.

Peripheral blood values: Except for HLA genotyping, Inaba et al. (2022) also collected peripheral blood samples at baseline and every visit during treatment. The rate of change in certain peripheral blood values was examined during the first 12 weeks of treatment, from baseline to the onset of the Type 1 DM irAE and during the 6 weeks before the onset of ICI-induced Type 1 DM. A statistically significant increase was observed in the absolute neutrophil count (ANC), the relative neutrophil count, the neutrophil–lymphocyte count, and the neutrophil–eosinophil rate when analyzing the rate of change during the 6 weeks before the onset. In the same statistical analysis, a statistically significant decrease in absolute lymphocyte count (ALC), absolute eosinophil count (AEC), and relative eosinophil count (REC) was demonstrated. However, a major limitation of the study, as the authors suggest, is the absence of comparison with a control group [56]. Consequently, further research is essential in order to confirm the utility of the above findings.

Autoantibodies: An attempt to associate the presence of autoantibodies and immune-related diabetes mellitus was conducted at two academic institutions in the US. Antibodies associated with diabetes mellitus (anti-GAD65, anti–ZnT8, anti–IA-2, islet cell antibody) were measured in 25 out of the 27 patients that developed diabetes mellitus following treatment with anti-PD-1 or anti-PD-L1. A control group of 12 patients with similar cancer types and treatment regimens was enrolled. At least one antibody was positive in 40% (10/25) of the cohort, compared to 25% (3/12) of the control group. Measurements of antibodies before and during treatment were conducted in three patients without clear conclusions [57]. Additionally, in their literature review, Lo Preiato et al. (2020) assessed the prevalence of diabetes-associated antibodies from available cases in their database. A total of 43.0% presented positive anti-glutamic acid decarboxylase (anti-GAD) antibodies, in contrast with the much higher prevalence of type 1 Diabetes Mellitus. No comparison was made with a control group [62]. Lastly, many case reports and case studies of immune-related Diabetes Mellitus have reported both positivity and negativity for specific diabetes-related antibodies [63,64,65,66,67]. However, no conclusion can be made without structured studies.

## 5. Endocrine irAEs

Although most studies focus on assessing predictive biomarkers for a specific endocrinopathy each time (e.g., biomarkers for thyroid dysfunction, biomarkers for hypophysitis, etc.), we found a reported association of markers with the development of endocrine dysfunction as a collective entity, without subcategorizing depending on the affected organ.

### 5.1. Pre-Treatment

Absolute eosinophilic count (AEC): In a study of 45 patients with advanced malignant melanoma treated with anti-PD-1 antibodies, the baseline absolute eosinophilic count was positively associated with the incidence of endocrine irAEs (*p* = 0.045). The ROC analysis determined a cut-off level of 240/μL to be a useful predictor for endocrine adverse reaction occurrence (sensitivity = 87.5%, specificity = 50%, *p* = 0.0134). The particular study did not analyze each type of endocrine adverse effect separately [51].

### 5.2. During-Treatment

Relative eosinophilic count (REC): Nakamura et al. also found a positive correlation between the relative eosinophilic count and endocrine irAE occurrence. The ROC analysis suggested that a cut-off value of REC at 1 month > 3.2% could be a useful biomarker to predict endocrine adverse reactions [51].

The above-mentioned immune-related endocrinopathies and their respective biomarkers can be found summarized, according to their appearance relative to treatment onset, in Table 1 and Table 2 and Figure 1.

## 6. Discussion

Immune checkpoint inhibitors are novel agents in the armamentarium of oncologic treatments. They provide a new approach to cancer therapy by reversing the tumor-induced immune suppression, thus enhancing the hosts’ immune response against cancer cells [68]. The widespread use of immunotherapy has substantially improved the prognosis and survival of several cancer types [69]. However, the resulting over-activation and decreased regulation of the immune system can trigger a specific subset of side effects, termed immune-related adverse events (irAEs). Immune-related adverse events are autoimmune disorders that can affect any organ in the body (Figure 2) [10]. Endocrine organs are frequently targeted and most commonly involve the pituitary and thyroid [22,23]. If detected promptly, most immune-related endocrinopathies can be adequately managed with hormone replacement or appropriate pharmacotherapy [21]. However, most symptoms are non-specific and require a high index of suspicion, or else they can be easily looked over. If the diagnosis is missed, it can lead to devastating, potentially life-threatening complications [22,69]. Thus, it is imperative to establish appropriate biomarkers in order to predict the development of immune-related endocrinopathies and monitor patients accordingly.

Predictive biomarkers can be subcategorized into two groups, pre-treatment and during-treatment biomarkers. Pre-treatment biomarkers estimate the individual risk of developing adverse effects from an immunotherapy agent prior to treatment initiation. The risk–benefit ratio can guide treatment selection for each patient, depending on his specific characteristics. Additionally, the potential risk dictates proper surveillance. The knowledge of increased risk for a specific side effect allows for close monitoring, early detection, and appropriate intervention. The patient-centered approach becomes even more efficient with the addition of biomarkers during treatment. It allows for a constant reevaluation of risk and indicates possible impending development of adverse effects, necessitating diagnostic evaluation.

Another point worth mentioning is the reported correlation of immune-related adverse effects with improved response to immunotherapy and higher survival rates [17,70,71,72,73,74]. In conjunction with the fact that most immune-related endocrinopathies can be successfully managed if diagnosed in a timely manner, it becomes clear that an increased risk of developing adverse effects should not rule out the use of a specific immunotherapy agent. Conversely, it might indicate increased benefit from treatment. However, as mentioned above, it necessitates adequate risk stratification of patients to assess the risk–benefit ratio and close monitoring for early detection and treatment.

Potential biomarkers that are being examined in the current literature include serum measurements such as hematologic indices, biochemical indices, cytokine assays, antibodies, and genetic polymorphisms. Radiographic and microbiology parameters are also being assessed. For example, the gut microbial signature has been associated with the development and severity of distinct irAEs as well as the outcome of anti-PD-1 immunotherapy [75,76]. Additionally, positron emission tomography with 18-F fludeoxyglucose integrated with computed tomography (18F-FDG PET/CT) can possibly detect select irAEs before clinical appearance and is considered a potential predictive modality [77]. However, these methods are still in the early stages of research and are not easily attainable in everyday clinical practice. In this review, we chose to concentrate on values from serum samples that are more established and accessible.

We identified possible reasons for not reaching clinical significance for the examined biomarkers. The incidence of immune-related endocrinopathies is estimated to be approximately 10%, with the incidence of each endocrine organ dysfunction being even lower [23]. Due to the low incidence, most studies did not have a sufficient cohort population to extract applicable information. Thus, studies with larger cohorts are needed in order to confirm proposed associations and transfer the existing findings into clinical practice. There are currently multiple ongoing clinical trials that aim to identify possible predictive biomarkers for immune-related toxicities and adjunctive characteristics, namely the MINER (Monitoring of Immunological Mechanisms and Biomarkers Underlying Efficacy and Toxicity of Cancer Immunotherapy) and MIRAE (Montreal Immune-Related Adverse Effects) studies as well as many others. The design of those clinical trials rely on collecting a variety of biological specimens (tissue biopsy, blood, plasma, saliva, stool) at baseline and regular intervals during treatment and searching for reoccurring patterns with cellular, immunologic, genetic, metabolic, and microbial analysis. In addition, although the pathophysiologic foundations of immunotherapy are largely common and rely on the activation of the immune response against tumor cells, different immunotherapeutic classes act via different mechanisms [68]. Subsequently, each class presents with different endocrine adverse effects or leads to the same endocrine adverse effect by a different pathway. This highlights the need for separate studies for each immune checkpoint inhibitor monotherapy treatment and combined regimens. It also becomes clear that a specific ICI-induced endocrinopathy might be associated with different biomarkers depending on the used therapeutic agent.

During this search, we encountered several obstacles in reviewing current data. Firstly, we found substantial inconsistencies in the terminology used. Immune checkpoint inhibitors are referred to as both ICI and ICPi. Terms such as “pituitary dysfunction”, “hypophysitis”, and “hypopituitarism” are mainly used interchangeably but sometimes as separate entities; “isolated adrenocorticotropin insufficiency” is similarly examined as a part of pituitary dysfunction or separately [43,78,79], and thyroid dysfunction may represent only overt presentations or also include subclinical instances [37,38,40]. This hinders adequate review of all existing data and the possibility of drawing universal conclusions. We call for the standardization of necessary terms and definitions. Lastly, we noticed an indistinct differentiation between predictive and diagnostic biomarkers. It is imperative to realize which parameters can be used to assess potential future risk for the development of side effects and which values should be utilized when an adverse effect is already suspected and needs to be confirmed. Predictive values can be subsequently incorporated into the selection of treatment regimens and appropriate monitoring modalities for each individual.

## 7. Conclusions

In the current review, we present an overview of proposed biomarkers for endocrine immune-related adverse effects in patients receiving immune checkpoint inhibitor therapy. We conclude that thus far, no biomarker has been proven to adequately predict the possibility of developing a specific endocrine adverse effect following immunotherapy. However, many associations have been made, and potential prospects are being examined. Most research has been conducted regarding pituitary and thyroid immune-related dysfunctions, which are the most commonly observed immune-related endocrinopathies in clinical practice. It is imperative to connect presumed associations with clinical utility. More studies with larger cohort numbers are needed in order to establish practical associations. The utilization of biomarkers promotes a patient-centered approach and should continuously be researched in order to improve provided health care services.

## Figures and Tables

**Figure 1 cancers-15-00375-f001:**
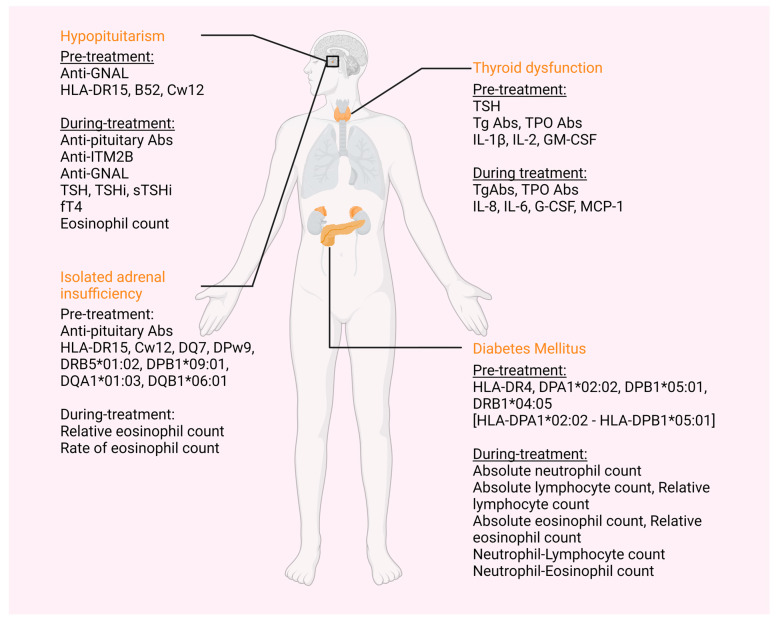
Immune-related adverse effects and corresponding possible biomarkers. Primary adrenal insufficiency and primary hypoparathyroidism have also been described following ICI treatment, but no biomarkers are available yet. HLA: human leukocyte antigen; Anti-GNAL abs: anti-guanine nucleotide-binding protein G(olf) subunit alpha antibodies; anti-ITM2B abs: anti-integral membrane protein 2B antibodies; TSH: stimulating thyroid hormone; TSHi: thyroid stimulating hormone index; sTSHi: standardized thyroid-stimulating hormone index; fT4: free T4; Tg Abs: thyroglobulin antibodies; TPO abs: thyroid peroxidase antibodies; IL: interleukin; G-CSF: granulocyte colony-stimulating factor; MCP-1: monocyte chemoattractant protein 1. (This figure was created based on the tools provided by Biorender.com, accessed on 7 November 2022).

**Figure 2 cancers-15-00375-f002:**
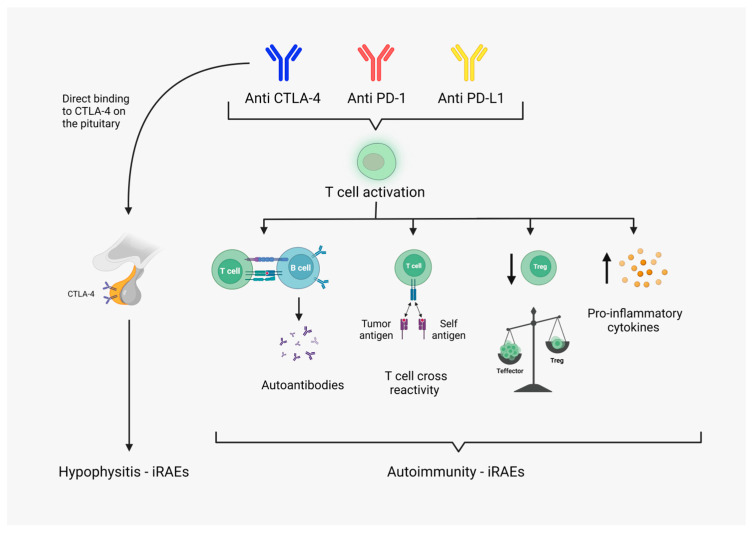
Possible pathophysiology of immune-related adverse effects: Immune checkpoint inhibitors induce T cell activation, thus enhancing the immune response to tumor cells. The resulting activation of cellular and humeral response is postulated to be the major cause of immune-related adverse effects. Non-specific activation of T cells leads to uncontrolled pro-inflammatory cytokine release and established inflammatory environment. T cell equilibrium leans toward effector T cells and downregulation of regulatory T cells. Suppression of Tregs results in loss of peripheral tolerance and subsequent auto-reactivity. Tumor-specific T cells cross-react with self-antigens on healthy cells. Overstimulation of T cells enhances T cell-B cell interaction and antibody production. Production of autoantibodies has been shown to mediate several organ-specific irAEs. Direct attack of anti-CTLA antibodies against physiologically expressed CTLA-4 on anterior pituitary cells. (This figure was created based on the tools provided by Biorender.com, accessed on 9 November 2022).

**Table 1 cancers-15-00375-t001:** Pre-treatment biomarkers for irAEs.

irAE	Biomarker	Biomarker Assessment	No of Papers
Thyroid	TSH	Higher levels	5 [28,29,30,31,32]
	TgAbs and/or TPOAbs	Presence	10 [28,29,31,32,33,34,35,36,37,38]
	IL-1β	Higher levels	1 [40]
	IL-2	Higher levels	1 [40]
	GM-CSF	Increase	1 [40]
Pituitary			
Hypopituitarism	Anti-GNAL Abs	Presence	1 [45]
	HLA-DR15	Presence	2 [42,43]
	HLA-B52	Presence	1 [42]
	HLA-Cw12	Presence	2 [42,43]
Isolated ACTH deficiency	Anti-pituitary Abs	Presence	1 [43]
	HLA-DR15	Presence	1 [43]
	HLA-Cw12	Presence	1 [43]
	HLA-DQ7	Presence	1 [43]
	HLA-DPw9	Presence	1 [43]
	HLA-DRB5*01:02	Presence	1 [44]
	HLA-DPB1*09:01	Presence	1 [44]
	HLA-DQA1*01:03	Presence	1 [44]
	HLA-DQB1*06:01	Presence	1 [44]
Diabetes Mellitus	HLA-DR4	Presence	1 [57]
	HLA-DPA1*02:02	Presence	1 [56]
	DPB1*05:01	Presence	1 [56]
	HLA-DRB1*04:05	Presence	5 [44,58,59,60,61]
	HLA-DPA1*02: 02-DPB1*05:01	Presence	1 [44]
Endocrine	Absolute Eosinophilic count	Higher levels	1 [51]

**Table 2 cancers-15-00375-t002:** During treatment biomarkers for irAEs.

irAE	Biomarker	Biomarker Assessment	No of Research
Thyroid	TgAbs and/or TPOAbs	Increase	3 [39,40,41]
	Thyroglobulin (Tg)	Elevation before third ICI treatment	1 [40]
	IL-8	Decrease	1 [40]
	IL-6	Increase	1 [39]
	G-CSF	Decrease	1 [40]
	MCP-1	Decrease	1 [40]
Pituitary			
Hypopituitarism	Anti-pituitary Abs	From negative pre-treatment to positive	2 [43,46]
	Anti-ITM2B	Increase	1 [45]
	Anti-GNAL Abs	Increase	1 [45]
	TSH	Decrease	1 [47]
	TSHi, sTSHi	Decrease	1 [48]
	fT4	Decrease	1 [48]
	Eosinophil count (/μL)	Increase	1 [49]
Isolated ACTH deficiency	Relative eosinophil count	Increase	1 [50]
	Rate of eosinophil count	Higher levels	1 [50]
Diabetes Mellitus	Absolute neutrophil count	Increase	1 [56]
	Relative neutrophil count	Increase	1 [56]
	Absolute lymphocyte count	Decrease	1 [56]
	Absolute eosinophil count	Decrease	1 [56]
	Relative eosinophil count	Decrease	1 [56]
	Neutrophil–Lymphocyte count	Increase	1 [56]
	Neutrophil–Eosinophil count	Increase	1 [56]
Endocrine	Relative Eosinophilic count	Higher levels	1 [51]

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
