# Peer review of "Predictive Biomarkers for Immune-Related Endocrinopathies following Immune Checkpoint Inhibitors Treatment"

_cancers, 2023, doi:10.3390/cancers15020375_

Round 1

Reviewer 1 Report

Dear Authors, 

congratulations for your significant effort. I must admit I really enjoyed reading the paper - not only it is very interesting and valuable, but also the language used makes it easy to digest and consolidate new information it provides. In my opinion it is crucial, especially for clinicians who don't have much free time, and yet, they must update their knowledge on a daily basis. Also, tables included are very useful both in order to find a proper information quickly and to sum up the information broadly described in the paragraphs. Figures are clear and informative, providing additional value to the revision.

I have no comments to add. Good luck, 

P.

Author Response

Reviewer 1

Dear Authors, 

congratulations for your significant effort. I must admit I really enjoyed reading the paper - not only it is very interesting and valuable, but also the language used makes it easy to digest and consolidate new information it provides. In my opinion it is crucial, especially for clinicians who don't have much free time, and yet, they must update their knowledge on a daily basis. Also, tables included are very useful both in order to find a proper information quickly and to sum up the information broadly described in the paragraphs. Figures are clear and informative, providing additional value to the revision.

I have no comments to add. Good luck, 

P.

AUTHOR RESPONSE: Dear reviewer, thank you very much for you encouraging and complementary comments. It is truly gratifying to have our hard work and efforts acknowledged by peers. Our goal was to deliver all current information in a comprehensive and concise form and we are glad our point got across.

We wish you the best of luck.

Reviewer 2 Report

Shalit et al has produced a very well written and cohesive review. Current literature is well covered and directions for future biomarker studies are outlined.

Minor comments:

1. line 600-601: Could you please elaborate?

2. Line 635: How do you define predictive vs diagnostic biomarkers? Every biomarker has the sense of prediction. Diagnosis is usually not based solely on biomarkers.

Author Response

Reviewer 2

Shalit et al has produced a very well written and cohesive review. Current literature is well covered and directions for future biomarker studies are outlined.

AUTHOR RESPONSE: Initially, we would like to thank the reviewer for his remarks and opportunity to elaborate on our findings. Below we provide additional information and our thought process in response to each comment to ensure the clarity of our statements.

  1. line 600-601: Could you please elaborate?

AUTHOR RESPONSE: As the medical community continues to progress and improve upon it self, we look for additional modalities to predict adverse effects. Such new methods in the field of irAEs are the microbiom identity of each patient and certain imaging modalities. However, these methods are still early in their development stage and well-established research studies are scarce. Additionally, although these methods provide a more individualized, patient-centered approach, we find they are not yet attainable in clinical practice. For these reasons, we decided to concentrate our review on serum measurements. To make our point more clear we added to our text the following segment:

‘Radiographic, and microbiology parameters are also being assessed. For example, the gut microbial signature has been associated with the development and severity of distinct irAEs as well as the outcome of anti-PD-1 immunotherapy [75,76]. Additionally, positron emission tomography with 18-F fludeoxyglucose integrated with computed tomography (18F-FDG PET/CT) can possibly detect select irAEs before clinical appearance and is considered as a potential predictive modality [77]. However, these methods are still in the early stages of research and not easily attainable in the everyday clinical practice. In this review, we chose to concentrate on values from serum samples that are more established and accessible.’

  1. Line 635: How do you define predictive vs diagnostic biomarkers? Every biomarker has the sense of prediction. Diagnosis is usually not based solely on biomarkers.

AUTHOR RESPONSE: We would like to thank the reviewer for his sharp remark. This differentiation did complex us at the beginning as well, which is why we made a point to mention it. Although the terms are seemingly similar, there is a distinct, fine differentiation. As mentioned in our review, predictive biomarkers are used to identify individuals that are at higher risk of developing a specific event – meaning that the individuals have not developed the adverse effect yet but are more likely to, in comparison with a similar individual without the specific biomarker. Diagnostic biomarkers are used to detect or confirm the presence of a disease or condition of interest (ie in our case, to confirm a certain irAE that has already developed in a patient). Although a diagnosis usually does not rely exclusively on a specific biomarker, we do use diagnostic biomarkers in clinical practice to confirm a diagnosis, in combination with the individuals’ symptoms, physical examination, imaging and general assessment.

To clarify our point we made a small modification to our explanation of the term predictive biomarker: “predictive biomarkers are used to predict if a certain individual or group of people is more likely to experience a specific event in comparison to similar individuals without the biomarker

We feel that this addition, combined with the analysis of our viewpoint in the linesline 635-638 delivers our message regarding the fine difference between the two categories of biomarkers.

Reviewer 3 Report

Congratulations! Generally the paper is really interesting and the information gathered for the review valuable and thorough. There are some things which should be checked and changed, which I list below, but overall it is a really valuable work written in very clear and concise English.

  • please unify irAEs abbreviation throughout the text  - sometimes it is irAEs and sometimes iRAEs and in other places irEAs and other variables;
  • line 143 - 145: those teams are cited in different order (Brilli - Luongo - Pollack) than listed in the text, please change the order for the sake of clarity;
  • line 246:
    • Diamandis et al. work is not on the references list;
    • there should be a proper citation - I mean a number of reference [40, 39];
  • line 273: this part mentions not only during-treatment markers but also after-treatment markers. Maybe it would be better to change a title of this part of the paper from 'During treatment' to 'During and after treatment'?
  • line 373:
    • please add the abbreviations You use in later pats of a paper (TSHi, sTSHi);
    • please unify the way You write fT4 (in some places it is fT4, in others FT4), the same goes for fT3;
  • lines 436 and 437: please use full names for those abbreviations;
  • 'The above data are summarized in Table 1, Table 2 and Figure 1.' fragment shuld be a separate part from the text in 2.4 paragraph as it summarizes all information from the paper;
  • line 611: there should be a full name mentioned of those studies.

The separate problems, editorial really, are:

  • the abbreviations listed below Figure 1. (line 532 - 538) should be a part of the figure's description on the same page;
  • try and format the tables so the one table is on one page only, it is much easier to analyse the content that way;
  • the text in lines: 568 - 576 should be a part of Figure 2. description, as it looks right now, it is a completely separate element.

Other, less important things have been marked in the pdf document I attach.

Again, congratulations! That is a great work.

Author Response

Reviewer 3

Congratulations! Generally the paper is really interesting and the information gathered for the review valuable and thorough. There are some things which should be checked and changed, which I list below, but overall it is a really valuable work written in very clear and concise English.

AUTHOR RESPONSE: Initially, we would like to thank the reviewer for his very in depth examination of our work and very constructive remarks that aim for the improvement of our manuscript. We would also like to thank him for his kind words that recognize our effort. Below we provide our detailed modifications in response to each comment to ensure the clarity of our statements.

Please unify irAEs abbreviation throughout the text  - sometimes it is irAEs and sometimes iRAEs and in other places irEAs and other variables; ( extra: Additional abbreviations mentioned on the PDF file and grammar mistake)

AUTHOR RESPONSE: We have made appropriate changes in the text for the term ‘irAEs’ and all the other abbreviations and points that were indicated. Thank you for pointing everything out. We appreciate your attention to detail.

line 143 - 145: those teams are cited in different order (Brilli - Luongo - Pollack) than listed in the text, please change the order for the sake of clarity;

AUTHOR RESPONSE: We thank the reviewer for his remarks. Indeed we made changes to the text to clarify the message and cited the articles in the correct order.

“ Specifically, Luongo et al. (2021) and Pollack et al. (2019) observed that baseline levels of TSH were significantly higher in patients that developed hypothyroidism, after ICI treatment compared to hyperthyroid and euthyroid patients (p<0.0025, p<0.05 respectively) [29–30], while Brilli et al. (2021) reported higher TSH baseline levels in overt thyroid dysfunction (p=0.003) [31].”

  1. Luongo, C.; Morra, R.; Gambale, C.; Porcelli, T.; Sessa, F.; Matano, E.; Damiano, V.; Klain, M.; Schlumberger, M.; Salvatore, D. Higher Baseline TSH Levels Predict Early Hypothyroidism during Cancer Immunotherapy. J. Endocrinol. Invest. 2021, 44, 1927–1933, doi:10.1007/s40618-021-01508-5.
  2. Pollack, R.M.; Kagan, M.; Lotem, M.; Dresner-Pollak, R. Baseline TSH Level Is Associated with Risk of Anti–PD-1–Induced Thyroid Dysfunction. Endocr. Pract. 2019, 25, 824–829, doi:10.4158/EP-2018-0472.
  3. Brilli, L.; Danielli, R.; Campanile, M.; Secchi, C.; Ciuoli, C.; Calabrò, L.; Pilli, T.; Cartocci, A.; Pacini, F.; Di Giacomo, A.M.; et al. Baseline Serum TSH Levels Predict the Absence of Thyroid Dysfunction in Cancer Patients Treated with Immunotherapy. J. Endocrinol. Invest. 2021, 44, 1719–1726, doi:10.1007/s40618-020-01480-6.

line 246:

Diamandis et al. work is not on the references list;

there should be a proper citation - I mean a number of reference [40, 39];

AUTHOR RESPONSE: Thank you for detecting the omission. The numerical citation was added to the main text. Diamantis was not the first author, which is why he is not visible in the reference list but the citation is accounted.

‘In addition to their role during baseline measurements, the appearance or elevation of thyroid antibodies during treatment could also signify the initiation of thyroid dysfunction [39-41].’

  1. Music, M.; Iafolla, M.; Soosaipillai, A.; Batruch, I.; Prassas, I.; Pintilie, M.; Hansen, A.R.; Bedard, P.L.; Lheureux, S.; Spreafico, A.; et al. Predicting Response and Toxicity to PD-1 Inhibition Using Serum Autoantibodies Identified from Immuno-Mass Spectrometry. F1000Research 2020, 9, doi:10.12688/f1000research.22715.1.

Full authors list:

‘Milena Music, Marco Iafolla, Antoninus Soosaipillai, Ihor Batruch, Ioannis Prassas, Melania Pintilie, Aaron R. Hansen, Philippe L. Bedard, Stephanie Lheureux, Anna Spreafico, Albiruni Abdul Razak, Lillian L. Siu, Eleftherios P. Diamandis’

line 273: this part mentions not only during-treatment markers but also after-treatment markers. Maybe it would be better to change a title of this part of the paper from 'During treatment' to 'During and after treatment'?

AUTHOR RESPONSE: Again, we would like to emphasize our gratitude to the reviewer. It is clear that he put time and effort to read our work his indications allow us to correct any small details that were not thoroughly explained. This comment was valuable in order to make the paper more comprehensive and complete. The markers were measured after the first treatment cycle, but during the treatment period. We clarified this point in the text:

“Kurimoto et al. (2020) examined the levels of IL-8 and MCP-1 before and 4 weeks after first ICI treatment (at the time of third ICI treatment) in their cohort.”

“Furthermore, Muir et al. (2022) measured the serum IL-6 levels before and after treatment initiation in order to examine a potential correlation with the occurrence of thyroid dysfunction.”

lines 436 and 437: please use full names for those abbreviations;

AUTHOR RESPONSE: The full terms were added.

'The above data are summarized in Table 1, Table 2 and Figure 1.' fragment shuld be a separate part from the text in 2.4 paragraph as it summarizes all information from the paper;

AUTHOR RESPONSE: This change was made.

line 611: there should be a full name mentioned of those studies.

AUTHOR RESPONSE: We thank the reviewer for rightly pointing out the necessity of providing the full names of the studies. Our goal is to allow the reader to search independently the information in our paper and develop a well-rounded opinion with all references available.

Provided now in the text: ‘the MINER (Monitoring of Immunological Mechanisms and Biomarkers Underlying Efficacy and Toxicity of Cancer Immunotherapy) and MIRAE (Montreal Immune-Related Adverse Effects) studies’

The separate problems, editorial really, are:

the abbreviations listed below Figure 1. (line 532 - 538) should be a part of the figure's description on the same page;

try and format the tables so the one table is on one page only, it is much easier to analyse the content that way;

the text in lines: 568 - 576 should be a part of Figure 2. description, as it looks right now, it is a completely separate element.

AUTHOR RESPONSE: The necessary editorial changes were made in order to make the paper more comprehensible.